# Blue Color Indices as a Reference for Remote Sensing of Black Sea Water

Evgeny Shybanov *, Anna Papkova, Elena Korchemkina  and Vyacheslav Suslin

Marine Hydrophysical Institute of the Russian Academy of Sciences, 2 Kapitanskaya St., 299011 Sevastopol, Russia
* Correspondence: e-shybanov@mail.ru

**Abstract:** In this paper, we propose to analyze the values of the "blue" color index for further use in additional atmospheric correction of Level 2 remote sensing reflectance data for the waters of the Black Sea. Regardless of seasonal phenomena, atmospheric conditions, and the type of water, the average color index in the short-wave region, according to in situ measurements CI(412/443), varies from 0.77 to 0.83. The most frequently observed value is 0.8. In turn, the values of the "blue" color index CI(412/443) according to the satellite data of MODIS Aqua/Terra, VIIRS SNPP, and OLCI Sentinel 3A scanners showed a large scatter in values based on the standard deviation of the sample. The paper proposes to introduce the value of the minimum allowable threshold CI(412/443) > 0.59 based on the small variance found from in situ measurements, as well as on the basis of a theoretical estimate of the possible values of the index CI(412/443) when varying the backscattering exponent and the exponent for the absorption approximation. The quality check of the remote sensing data showed that, according to this selection criterion, 15% of data are physically incorrect for MODIS Aqua, 30% for MODIS Terra, 20% for Sentinel 3A, and 26% for VIIRS SNPP. In the course of the work, it was shown that the MODIS Aqua satellite provides the most high-quality and reliable information about the optical characteristics of the Black Sea.

**Keywords:** remote sensing reflectance; color index; short-wave spectrum; atmospheric correction; Black Sea

## 1. Introduction

The values of the remote sensing reflectance Rrs(λ) in the "blue" region of the spectrum (400–443 nm) are subject to large uncertainties in coastal and inland waters, usually referred to as Case 2 waters (based on the classification presented in [1]) [2–7]. For areas of such water, statistical relationships between the chlorophyll-a concentration (Chl-a) and colored dissolved organic matter (CDOM) are weak, while the ratio of optically significant material concentrations is strongly shifted towards nonliving organic matter—detritus and CDOM. This is explained by the fact that, for Case 2 water areas, in addition to CDOM of autochthonous origin, a large amount of CDOM enters the sea with various coastal runoffs. Therefore, for such areas, the separation of light absorption by phytoplankton from light absorption by nonliving organics is very difficult, which can result in poor-quality satellite products. In addition to the complex assessment of the bio-optical properties of the marine environment, the short-wave region is negatively affected by incorrect consideration of atmospheric influences, such as dust aerosol, smog, and industrial aerosol emissions [2]. Previously, using the Black Sea as an example, it was demonstrated that dust aerosol leads to obvious systematic errors in the retrieved Rrs(λ) values, which are characterized by negative values in the spectral range of 400–443 nm [8–12]. A similar effect was noted in the case of the presence of burning biomass over the region [12–14]. An interesting fact is that, even in the presence of a fine fraction of the background aerosol, the extrapolation error of the aerosol scattering at the wavelength λ is proportional to the second-degree polynomial of the wavenumber $k = 2\pi/\lambda$. This effect is explained by inaccurate estimates

of the contribution of the fine fraction of aerosol particles to the radiation scattered by the atmosphere [15]. An additional difficulty of measuring Rrs($\lambda$) in the short-wave region is that uncertainties are increased with the proportion of scattered radiation, solar zenith angles, and viewing angles [16].

Nevertheless, with the involvement of additional information from the "blue" region of the spectrum, the efficiency of using space observations can be significantly higher [17]. For example, using remote sensing reflectance values at 412 nm should be more efficient for separating the absorption by CDOM and by phytoplankton than is currently possible in the remote sensing paradigm [18]. As O'Reilly and Werdell noted in the problems of the Ocean Color, the UV band (412 nm) has rarely been used in empirical algorithms. While the peak of the specific absorption of chlorophyll coincides with the 443 nm band present in most ocean color sensors, the value of absorption at 412 nm specific for chlorophyll can reach more than ~70% of the value at 443 nm. Almost a third of the total absorption of chlorophyll between 400 and 700 nm falls below 443 nm, which suggests that bands below 443 nm, such as 412 nm, can also be useful for the detection of chlorophyll with certain conditions and assumptions. Therefore, the 412 nm band was used in two new OC5 and OC6 algorithms [19]. Additionally, with reference to the Rrs($\lambda$) in the short-wavelength region, it is possible to build a fundamentally new or additional algorithm to eliminate the atmosphere influence by interpolation methods. In order to carry out the atmospheric correction procedure, it is necessary to set the value of Rrs($\lambda$) in the short-wavelength region. For the Black Sea, the following parametrization methods have been proposed: neglecting the values of Rrs($\lambda$) in the near ultraviolet and constancy of values in the "blue" short-wavelength region [15], estimating the values of Rrs(412) from the condition that the corrected spectrum at 412 nm is close to the model spectrum described by two parameters [20,21].

In this paper, we propose to use the values of the "blue" color index for the additional correction of remote sensing optical data for the Black Sea waters. The "blue" color index is the ratio of the remote sensing reflectance in two bands in the short-wave region, for example, Rrs(412)/Rrs(443), Rrs(400)/Rrs(443), and Rrs(400)/Rrs(412) ratios. In scientific research, band differences are often used under the term "color index" [22–24]. These algorithms are aimed at eliminating atmospheric correction errors and use three channels to do this in a wide range of wavelengths, for example, blue–green–red [22]; 490, 550, and 670 nm [23], and 547, 667, and 748 nm [24]. The Rrs values in the extreme channels are interpolated to the middle of the spectral range. After that, the difference is calculated, which characterizes the value of the backscattering coefficient. In this article, we do not analyze the features of our own atmospheric correction algorithm but only will consider in detail the ratios Rrs(412)/Rrs(443) and Rrs(400)/Rrs(443), further denoted as the color indices CI(412/443) and CI(400/443). It is assumed that the values of the "blue" color index for coastal waters, where CDOM determines the main bio-optical properties of the sea, cannot vary widely. The index constancy condition can be further used in new algorithms for additional atmospheric correction.

It is worth noting that there are already some algorithms that use the "blue" color index. Morel and Gentili developed a correction that can be applied to OC4-type of algorithms to account for the deviations in absorption by CDOM from the Case 1 waters [25]. This approach is based on the assumption that the ratio of reflectance at 412 nm to that at 443 nm is mainly sensitive to CDOM, albeit influenced to some extent by chlorophyll, and that the ratio of reflectance at 490 nm to that at 555 nm is essentially dependent on chlorophyll, although also influenced to some extent by CDOM. This approach uses the bio-optical model of Morel and Maritorena [26], which implicitly includes a prescribed relationship between CDOM absorption and chlorophyll, and thus produces a unique set of curves relating Rrs(412)/Rrs(443) to Rrs(490)/Rrs(555). Deviations from the prescribed relationship are introduced using a factor $\phi$, with $\phi > 1$ indicating an excess and $\phi < 1$ a deficit of per unit chlorophyll. Morel and Gentili produced a 2D look-up table relating Rrs (412)/Rrs (443) to Rrs(490)/Rrs(555) for specific discrete values of $\phi$. Relative anomalies

in CDOM (ϕ) concerning its standard (chlorophyll-related) values can then be computed efficiently using reflectance ratios derived from ocean color. The operation of this algorithm has been verified for the water area of the Mediterranean Sea in [27] using in situ and SeaWiFS (Sea-viewing Wide Field-of-view Sensor) satellite measurements; it was shown that Rrs(412)/Rrs(443) varies from 0.8 to 1.1, while the Rrs(490)/Rrs(555) ratio varied from 0.8 to 4.5. It was found that the level of CDOM in the Mediterranean Sea is about twice as high as in nearby oceanic waters with similar trophic conditions and has shown a pronounced seasonal variability. In this study, the insufficient spatial–temporal coverage by in situ measurements was noted and marked as a separate issue.

Similar results were obtained in [28] when analyzing all possible data in the NASA bio-Optical Marine Algorithm Dataset (NOMAD). It was calculated that the Rrs(412)/Rrs(443) value varied from 0.65 to 1.2. Lee and Hu documented that the average reflectance ratio for Case 1 seawaters is Rrs(412)/Rrs(443) = 1.2 ± 0.2 [29]. Later, using the data for the coastal zone of the Beaufort Sea (Case 2 waters), it was demonstrated that the entire data set, in contrast, falls well below the average value and goes beyond ± 10% variability, indicating a much lower than expected Rrs(412)/Rrs(443) [30]. This is considered a clear sign of excessive CDOM absorption.

The weak variability of the "blue" color index was also noted in work related to regional ocean color chlorophyll algorithms for the Red Sea [31]. Using synchronous pairs of in situ data and satellite measurements from the MODIS Aqua, it was found that the CI(412/443) for the Red Sea varies from 0.6 to 1.1. Moreover, the study noted the seasonal dependence of the variation of the "blue" color index; in March 2013, CI(412/443) changed from 0.6 to 0.9, and in November, from 0.9 to 1.1. Presumably, this phenomenon is caused by the seasonal phytoplankton bloom characteristic of the spring period in the Red Sea.

This "blue" band ratio was also used in this work [7]. This study developed an algorithm to estimate Rrs(41×) (410 or 412) and Rrs(443) when the satellite Rrs($\lambda$) in blue bands suffer from large uncertainties. The algorithm first determines the Rrs($\lambda$) spectral shape from the satellite-measured Rrs($\lambda$) values at three wavelengths of 48× (486, 488, or 490), 55× (547, 551, or 555), and 67× (667, 670, or 671) nm. The algorithm then derives Rrs(41×) and Rrs(443) from the estimated Rrs($\lambda$) spectral shape with algebraic equations. The efficiency of this algorithm was confirmed by validating satellite measurements (SeaWiFS, MODIS Aqua, and VIIRS-SNPP) and in situ Rrs ($\lambda$) matchups from global waters. It was shown that the uncertainties of the estimated Rrs(41×) and Rrs(443) are substantially smaller than the ones of the original satellite products. All of the data from the SeaBASS validation database were used as input data, and the CI(412/443) ratios (in the case of VIIRS CI(410/443)) were calculated. It was found that these ratios for in situ measurements (look-up tables [32,33]) ranged from 0.6 to 1.4, while the satellite products had a much larger scatter. Namely, for the SeaWiFS data, they varied from 0.2 to 1.6, and from VIIRS and MODIS Aqua data, CI(410/443) ranged from 0.1 to 1.1. It should be noted that the highest concentration of all studied ratios was within the range of 0.6 to 1.0.

Previously, in [34], a regional algorithm for additional correction of Level 2 satellite Rrs($\lambda$) was presented based on the constancy of the "blue" color index. The new method showed high efficiency when comparing satellite, model, and in situ measurements, even when a fine fraction of background and arid aerosols was present. It was assumed that the retrieved values of Rrs($\lambda$) can be calculated using the following equation:

$$\text{Rrs}_m(\lambda) = \text{Rrs}_{sat}(\lambda) + k \cdot \lambda^{-4} \tag{1}$$

Since the contribution of the aerosol component has the greatest influence on the short-wavelength region, it was proposed to calculate the *k* value based on the "blue" color index as

$$k = \frac{CI\left(\frac{412}{443}\right)\text{Rrs}_{sat}(443) - \text{Rrs}_{sat}(443)}{412^{-4} - CI\left(\frac{412}{443}\right)443^{-4}} \tag{2}$$

Using daily average in situ measurements from the AERONET-OC platforms from 2011 to 2021, it was demonstrated that the value of the color index CI(412/443) is 0.8 for the Black Sea, with a relative error of about 10%. This result is somewhat unexpected since the reflectance coefficient of the sea depends on absorption and scattering by optically significant materials, and their concentrations in coastal waters vary in a wide range. The quasi-constancy of the "blue" color index can be explained by the asymptotic behavior of IOPs (inherent optical characteristics) of seawater due to several causes. The first cause is associated with an increase in hydrosol concentrations, which leads to a nonlinear dependence of scattering on concentration. For example, at concentrations typical of coastal waters, the concentration dependence weakens and reaches saturation at values higher than $100 \text{ g/m}^3$ [35]. The second cause is the aggregation of particles with an incredibly large proportion of organic molecules [36–38]. The third cause is the smoothing of the shape of the phytoplankton absorption spectrum [39]. At the same time, the presence of certain correlations between the concentrations of optically significant materials (suspended mineral particles, phytoplankton, CDOM) cannot be ruled out. In this regard, the CI(400/412) index is expected to be more stable as the contribution of phytoplankton absorption decreases. The index value shows the spectral properties of Rrs($\lambda$), namely, the spectral slope in the selected region of the spectrum. Therefore, the measure of the uncertainty of using the "blue" color index in atmospheric correction procedures will be the value of the normalized standard deviation, i.e., $SD(\lambda_1/\lambda_2)/(\lambda_2 - \lambda_1)$, where $\lambda_1$, $\lambda_2$ are the wavelengths.

This paper proposes a detailed analysis of the variability of the "blue" color indices for an extended set of in situ data, which includes not only measurements from AERONET-OC stations from 2011 to 2022 (level 2) but also field data for the central and northeastern parts of the Black Sea. The purpose of this analysis is to confirm the low variability of the "blue" color index based on a larger dataset, which will then be compared with remote sensing data. It is worth noting that, in [35], an intermediate level of quality (level 1.5) was used for AERONET-OC in situ measurements of normalized water-leaving radiance. This paper considers data from AERONET-OC platforms of the highest quality optically significant materials—level 2 (more details provided in Section 2). An additional difference from the analysis of the variability of the "blue" color index from the previous work is the use of three AERONET-OC stations rather than two (measurements from the Section-7_Platform were added). In contrast to [35], a larger set of the "blue" color index possible values were applied using not only the 412 nm and 443 nm channels but also 400 nm. Additionally, as an illustration, the paper compares similar ratios for modern color scanners, such as MODIS Aqua/Terra, VIIRS SNPP, and Sentinel OLCI. The main goal of the work is to develop a remote sensing reflectance quality criterion for level 2 data based on the value of the "blue" color index. The obtained estimates of the remote sensing reflectance variability in the blue region will serve as an additional criterion for the quality of satellite and in situ data.

## 2. Materials and Methods

### 2.1. In Situ Measurements of Rrs($\lambda$)

For the Black Sea, an array of in situ measurements is presented both as data from two AERONET-OC platforms located mainly in the northwestern part and also as expeditionary data collected by employees of the MHI RAS using a spectrophotometer developed at the institute. Station positions, oceanographic platforms, and bathymetry are given in Figure 1.

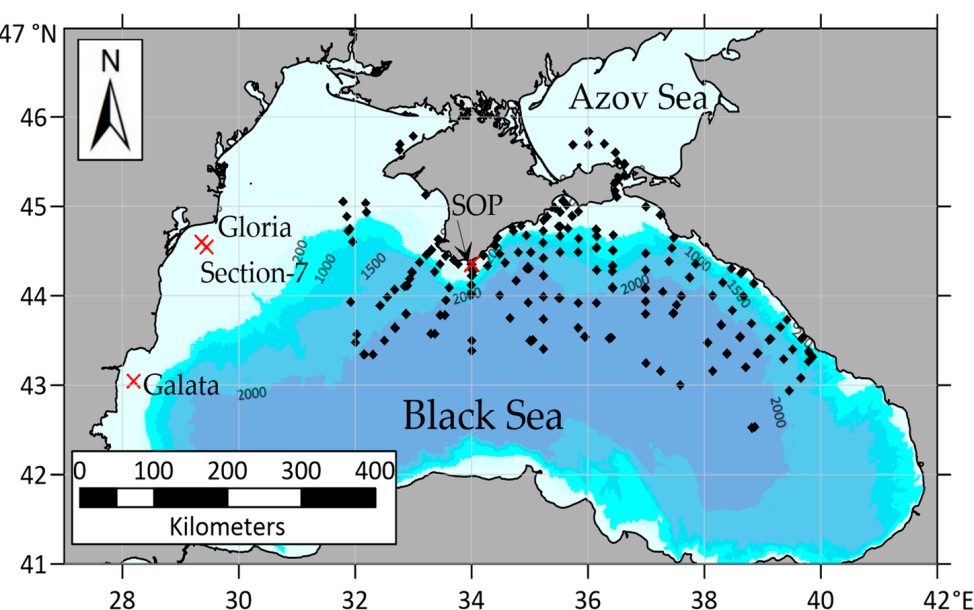

**Figure 1.** Optical stations made during the cruises (black dots); oceanographic platforms (red crosses).

2.1.1. In Situ MHI RAS Measurements

The measurements of Rrs($\lambda$) were held in 2002–2021 on the stationary oceanographic platform (SOP) of the Black Sea Hydrophysical Polygon near the south coast of Crimea (44.39N, 33.98E) as well as during six research cruises of R/V "Professor Vodyanitsky" in 2019–2021. The amount of data and their seasonal distribution are shown in Table 1.

The stationary oceanographic platform (SOP) of the Black Sea Hydrophysical Polygon near the south coast of Crimea is located in the sea 600 m off the shore. The depth of the sea in the installation area is 26–30 m. The main purpose of the oceanographic platform is to provide scientific in situ studies of the marine environment.

*a.* *Measurement system*

To obtain Rrs($\lambda$), it is necessary to measure the water upwelling radiance $L_u$ and the radiance of the sky reflected by the water surface $L_r$ [40]. For these measurements, a single spectrometer was utilized with an additional band for measuring downwelling irradiance, as described in [21,41]. The MHI spectrometer is designed to directly measure the reflectance coefficient. The operation implements a two-beam scheme, which allows the instrument to simultaneously measure the noncalibrated upwelling radiance (of the sea, white plate, or cuvette) and the noncalibrated downwelling irradiance [40]. The instrument has a turning objective lens for the lower hemisphere and a collector plate of milky glass on top. The downwelling radiation that has passed through the milk glass is reflected by a beam splitter plate to the monochromator window. Upwelling radiation, meanwhile, passes directly through the beam splitter. The cylindrical shutter of the rotating obturator successively blocks two light streams. Thus, both signals are recorded by one photosensitive element, and the level of the light signal from the collector, being reduced by an order of magnitude, becomes comparable to the upwelling radiance. Thus the raw output is a ratio of both signals with the value proportional to the radiance reflectance. The coefficient of proportionality is found by comparing the reflection data of the measured standard (white plate) with their tabulated values. The spectral range of the spectrometer is 390–750 nm with 1 nm step, and the maximum measurement error does not exceed 3%.

In addition to using the spectrophotometer itself, radiance reflectance measurements involved the use of a specially designed cuvette and a white calibration screen. The cuvette was designed to measure sky radiation reflected by the water's surface. To do this, it was filled with clean filtered water up to about 5 cm. The cuvette size was 200 mm × 100 mm length × width. Its walls and bottom were made of 3 mm neutral dark glass that absorbs

>99% of the light [42]. The reflection coefficient from the dark glass in the water was estimated at 0.004. To avoid the first order of reflection from the bottom of the cuvette, it is sufficient to tilt the bottom by 15 degrees, as shown in Figure 2b. The main cause for using the absorbing dark cuvette was to take into account polarization effects without calculating them. In other cases, the errors produced by polarization effects can reach dozens of percent (see the full explanation in Appendix A). It should be noted that when using this technique and observing the sea at an angle of 30°, the error of not taking into account waves at a wind speed of 5 m/s would be 5.1%. To calculate this uncertainty error, the expression $R_f(\theta) = (R_f(\theta + SD) + R_f(\theta - SD))/2$ was used, where $R_f$—is the Fresnel coefficient and $\theta$ is the angle of observation, and $SD$ is the standard deviation of wave slopes according to Cox and Munk [43].

**Table 1.** In situ $Rrs(\lambda)$ data description.

| | SOP Measurements | | | Cruise Measurements | |
|---|---|---|---|---|---|
| **Year** | **Dates** | **Amount** | **Year** | **Dates** | **Amount** |
| 2002 | 28 July–15 August | 18 | 2019 | 19 April–11 May | 101 |
| 2003 | 16–29 July | 41 | 2019 | 12–20 October | 10 |
| 2004 | 31 August–13 September | 41 | 2020 | 20 September 20–5 October | 7 |
| 2007 | 8–21 July | 70 | 2021 | 22 April–15 May | 85 |
| 2007 | 4–12 October | 38 | 2021 | 30 July 30–7 | 18 |
| 2008 | 10–13 September | 21 | 2021 | 3–18 September | 18 |
| 2010 | 11–16 August | 35 | | | |
| 2010 | 23–28 September | 30 | | | |
| 2012 | 7–16 July | 72 | | | |
| 2014 | 11–14 August | 19 | | | |
| 2015 | 16–24 September | 29 | | | |
| 2016 | 20–30 September | 25 | | | |
| 2017 | 24–31 May | 27 | | | |
| 2017 | 4–11 October | 27 | | | |
| 2018 | 29 September–9 October | 31 | | | |
| 2019 | 21–27 June | 16 | | | |
| 2021 | 25 June–8 July | 20 | | | |

*b.  Observation Geometry and Observation Conditions*

The upwelling radiance reflectance was measured above the water surface from the board of the R/V or SOP. All in situ measurements were made during daylight hours. Measurements were carried out under the following weather conditions: (i) cloudless atmosphere, or (ii) in the presence of clouds, if clouds were absent both near the Sun and at a point in the sky corresponding to its specular reflection from the sea surface, or (iii) with a uniform overcast sky, and (iv) wind speed did not exceed 5 m/s.

The device (spectrophotometer) was located on the bow of the ship, or in case of the platform, on the outrigger bridge. Their heights above sea level were 5 and 12 m, respectively. Typically, the sensor was directed at 30° to the nadir, and the azimuth relative to the sun was 90°. The observation geometry could change since the main criterion for choosing the direction of sighting was the absence in the field of view of (i) shadows from the vessel (platform) structures, (ii) cloud reflections, and (iii) floating objects.

Two consecutive measurements with the same measurement geometry were performed: (i) the total sea reflectance $\rho_u = L_u/E_d$ above the water surface, which can be expressed as $\rho_w + \rho_r = L_w/E_d + L_r/E_d$ (Figure 2a); and (ii) the reflectance $\rho_{cuv}$ of the absorbing cuvette filled with water (Figure 2b). Notice that the upwelling underwater radiance of a ~5 cm layer of water was negligible, so $L_{cuv} \approx L_r$. Each measurement took ~5 min. This short time interval lets us assume that the illumination conditions do not change during measurements.

### c. Calibration

Figure 2c illustrates the calibration procedure with a white standard Spectralon® panel, which is supposed to be Lambertian. To solve the main problems of this study on the values of "blue" color indices, the most important variable is the shape of the reflection spectrum of the sample (reference), not the absolute values. Let $\rho^*$ be the raw value of reflectance, then the calibration factor is $R_p/\rho_p^*$, where $R_p$ is a known panel reflectance. The measured reflectance values are the product of the calibration factor and raw data.

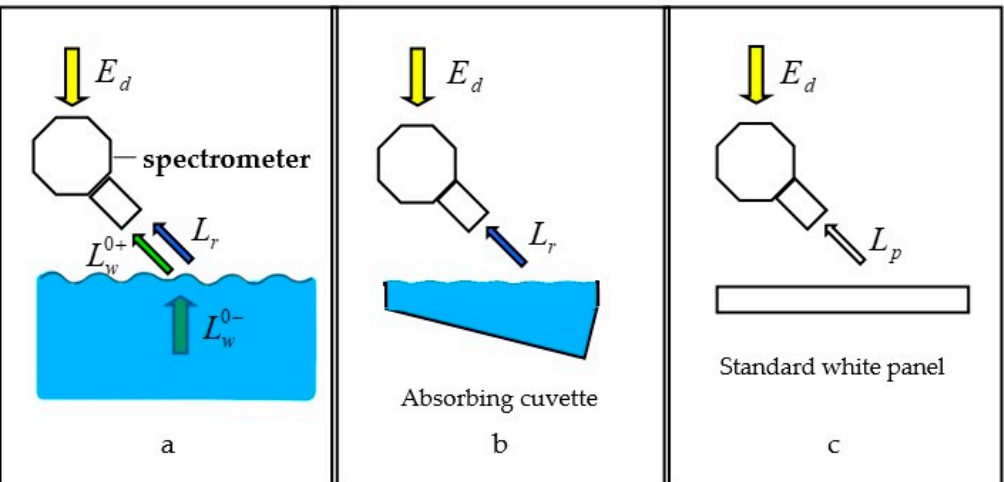

**Figure 2.** The measurement scheme: (**a**) the total upwelling reflectance, (**b**) the surface reflectance, and (**c**) reflectance of the white panel.

### d. Data Processing

After measurements, the spectra were smoothed by the median filter, and the remote sensed reflectance spectra Rrs were calculated according to [44]:

$$\text{Rrs} = \frac{1}{\pi}\frac{\rho_u^* - \rho_{cuv}^*}{\rho_p^*}R_p = \frac{1}{\pi}\frac{L_u - L_{cuv}}{L_p}R_p = \frac{1}{\pi}\frac{L_w}{L_L}, \tag{3}$$

where $L_L = L_p/R_p$ is the radiance of Lambertian surface.

We do not take into account the bidirectional reflectance distribution function for several causes: (i) we do not normalize the radiances and make measurements at azimuth near 90° relative to the sun, where bidirectional factor f/Q have the least angular dependence; (ii) the spectral variation of f/Q in the range 412–443 is less than 3% for coastal waters where Chl-a = 0.3–1 mg/m$^3$ [45]; (iii) the color indices are the ratios of reflectance coefficients at close wavelengths, so the BRDF is canceled out.

### e. Data Insurance and uncertainties

Two reliability criteria are used to check the quality of in situ data. The first one is the visual (rough) criterion for the Black Sea. It is known that in most cases, the correct spectrum has one local maximum in the middle of the visible region. In the near IR, the Rrs curve decreases monotonically to values approximately 30 times smaller than the Rrs value at its maximum. The second one is that all measurements containing negative values in the visible range of the spectrum are also excluded. It should be noted that minor small negatives in the NIR exceeding minus 0.0001 are allowed.

The main sources of errors are the conditions of observation, i.e., the state of the sea surface and the variability of the atmosphere. Instrumental sources of errors include the imperfection of the irradiance collector (non-cosine milk glass). Additionally, the possible errors of this method can be caused by the difference in the state of the water surface in

the sea and the cuvette. In case of high surface roughness in the sea, the measurements are repeated several times and then averaged.

This method is consistent with the NASA protocols [46]. The proposed method was used in sea expeditions [47] or from a stationary oceanographic platform [41]. The obtained spectra will be further denoted as in situ MHI spectra.

### 2.1.2. AERONET-OC Measurements

One of the most effective sources of studying the characteristics of atmospheric aerosol, as well as in situ measurements of ocean color, is the global network of observational ground-based automated stations (platforms) AERONET (Aerosol RObotiesNETwork). The advantage of this network is the use of the same type of automatic photometers (Cimel-318) and standardized procedures for calibration and processing of the received data. The AERONET network, mostly developed for atmospheric research at various scales, has been expanded to support marine applications. This new network component, called AERONET-Ocean Color (AERONET-OC), provides an additional capability to measure the water-leaving radiance. AERONET-OC plays an important role in ocean color satellite activities through standardized measurements that are (a) performed at different locations using a single measurement system and protocol, (b) calibrated using an identical reference source and method, and c) processed using the same code [48]. At the moment, only two Black Sea stations provide information on the ocean color according to the measurements of Section-7_Platform (29.45°E, 44.45°N) (in the past: Gloria) and Galata_Platform (28.19°E, 43.05°N) stations. Section-7_Platform AERONET-OC (29.36°E, 44.60°N) is located approximately 12 nautical miles off the Romanian coast south of the Danube Estuary; the water depth there is about 40 m. The Galata_Platform is located approximately 13 nautical miles off the coast of Bulgaria, in front of the city of Varna. The water depth in this area is 35 m. For the western part of the Black Sea, data on the water-leaving radiance ($L_W$) are regularly provided, as well as the normalized water-leaving radiance ($L_{WN}$) calculated by the method proposed by Zibordi et al. [49] to remove the dependence on survey geometry and bidirectional effects in $L_W$. It is worth noting since, in the future, satellite and in situ measurements of the water-leaving radiance will be validated, and all values of $L_{WN}(\lambda)$ will subsequently be converted to Rrs($\lambda$) by dividing by the solar constant Fo($\lambda$):

$$Rrs(\lambda) = L_{WN}(\lambda)/Fo(\lambda). \qquad (4)$$

In this work, we used all openly available data from the AERONET-OC stations listed above from 2010 to 2022 at quality Level 2, that is, with the use of qualitative atmospheric correction [48]. It should be noted that from 2010 to 2018, measurements were carried out at 412 nm, 443 nm, 490 nm, 532 nm, 551 nm, and 667 nm. Starting in 2018, the band set has changed to 400 nm, 412 nm, 443 nm, 490 nm, 510 nm, 560 nm, and 667 nm. In this work, we used daily average data of the Level 2 normalized sea radiance $L_{WN}$, which is considered to have better quality. Level 1.5 includes only cloudiness screening using a series of quality tests, while Level 2 consists of completely cleaned data obtained after calibration and software verification [49]. In the course of the research, the values of $L_{WN}(\lambda)$ were converted into Rrs($\lambda$) by dividing by the solar constant Fo($\lambda$) [50]. Since the stations in the Black Sea are located near the coast at the same distance, at almost equal depth, and affected by the same river runoff, it was decided not to separate them for further data analysis. So the array of measurements has 1412 average daily values from 2010 to 2018 and 1208 values from 2018 to 2022.

### 2.2. Remote Sensing Measurements

The source of Rrs($\lambda$) satellite measurements was the MODIS Aqua/Terra, VIIRS SNPP, and Sentinel 3A spectroradiometers. Later, on the basis of these values, the satellite values of the color index in the blue region of the spectrum were calculated.

The MODIS (Moderate Resolution Imaging Spectroradiometer) spectroradiometer is one of the key instruments aboard the US Terra and Aqua satellites of the EOS series.

MODIS has 36 spectral bands with 12-bit radiometric resolution in the visible, near, mid, and infrared ranges. Due to the continuous mode of operation and a wide survey swath (2330 km), any territory within the station's visibility zone is observed at least once a day. This provides the use of MODIS data for solving various problems of regular monitoring of natural phenomena within a large region (control of ice conditions, observation of snow cover dynamics, monitoring of forest fires, flood situation, the state of crops in agricultural fields, etc.) [51]. Remote sensing reflectance (Rrs($\lambda$)) (sr$^{-1}$) is determined for spectral bands 412, 443, 469, 488, 531, 547, 555, 645, 667, and 678 nm. The concentration of chlorophyll a, (mg m$^{-3}$), according to MODIS data, is calculated using Rrs($\lambda$) values for 2–4 wavelengths from the range of 440–670 nm.

As a follow-up, another instrument is used, specifically VIIRS SNPP. A primary sensor onboard the Suomi-National Polar-orbiting Partnership (SNPP) spacecraft, the Visible Infrared Imaging Radiometer Suite (VIIRS) has 22 bands: 14 reflective solar bands (RSBs), 7 thermal emissive bands (TEBs) and a Day Night Band (DNB). The RSBs cover the spectral wavelengths between 0.412 to 2.25 um and have three (I1-I3) 371 m and eleven (M1-M11) 742 m spatial resolution bands. The VIIRS data provide better resolution of satellite images in contrast to MODIS and represent a different set of bands; in the short-wave region, the bands 410 nm, 443 nm, and 486 nm are used [52].

The most advanced and accurate remote sensing instrument currently available is The Ocean and Land Color Instrument (OLCI) installed aboard Sentinel 3A, and Sentinel 3B is a European Space Agency Earth observation satellite dedicated to oceanography that launched on 16 February 2016. The Ocean and Land Color Instrument (OLCI) is the successor to ENVISAT's Medium Resolution Imaging Spectrometer (MERIS), having additional spectral bands, different camera arrangements, and simplified onboard processing. The OLCI is a push-broom instrument with five camera modules sharing the field of view. The field of view of the five cameras is arranged in a fan-shaped configuration in the vertical plane, perpendicular to the platform velocity. Each camera has an individual field of view of 14.2° and a 0.6° overlap with its neighbors. The whole field of view is shifted across the track by 12.6° away from the sun to minimize the impact of sun glint. OLCI is equipped with onboard calibration hardware based on sun diffusers. There are three sun diffusers: two "white" diffusers for radiometric and one for spectral calibration, with spectral reflectance features. The native resolution is approximately 300 m, referred to as Full Resolution (FR). A Reduced Resolution (RR) processing mode provides Level-1B data at sampling rates decreased by a factor of four in both spatial dimensions resulting in a resolution of approximately 1.2 km [53]. The main objective of the Sentinel-3 mission is to measure sea surface topography, sea and land surface temperature, and ocean and land surface color with high accuracy and reliability to support ocean forecasting systems, environmental monitoring, and climate monitoring. The Sentinel-3 Mission Guide provides a high-level description of the mission objectives, satellite description, and ground segment. It also covers an introduction to heritage missions, thematic areas and services, orbit characteristics and coverage, instrument payloads, and data products [54]. The wavelengths of Ocean Color measurements for this satellite in the blue region of the spectrum coincide with the MODIS bands. All satellite data are not copyrighted and are distributed freely.

Data from both the AERONET-OC stations and the SOP station were used in the validation of satellite data. For comparison with Ocean Color data on the location of AERONET-OC stations, we used the SeaBASS (SeaWiFS Bio-optical Archive and Storage System) database, which provides the best comparison between each satellite observation and the corresponding in situ measurement in time and proximity of points [55]. Satellite measurements were obtained from a rectangle of pixels (5 × 5) centered over the in situ measurement site. The data entering the SeaBASS database are cleared of all pixels containing the following error flags: ground (LAND), stray light (STRAYLIGHT, HIGLINT, HILT, ATMWAR), L$_W$(555 nm) < 0.15 (LOWLW), navigation errors (NAVFAILE), cloud boundaries or ice (CLDICE). Additionally, all spectra containing negative values of Rrs($\lambda$) in the short-wave region were excluded from the analysis since this is an indicator of

atmospheric correction errors, and such measurements have no physical meaning. For the VIIRS SNPP data, the 410 nm band, which is standard for this instrument, was used instead of 412 nm.

For comparative analysis of the satellite and SOP-measured remote sensing reflectance, measurements obtained from the OLCI satellite were used. The use of OLCI [56] satellite products for developing regional algorithms or checking the quality of atmospheric correction has obvious advantages over MODIS-Aqua/Terra [57,58] satellite products for comparison with in situ measurements performed on an oceanographic platform located at a distance of 600 m from the shore. These advantages include different spatial resolutions (300 m and 1000 m), which can affect the stray illumination of a pixel (STRAYLIGHT) [59], minimization of the effect of spatial inhomogeneity of the optical properties of the upper water layer in the coastal area of the sea, the presence of a spectral band with the central wavelength of 400 nm and, finally, the service life of the equipment, which for MODIS-Aqua/Terra is already close to completion. Since the OLCI data have been available since 2016, the data from the Katsiveli platform were also analyzed in 2016–2021 period (Table 1).

## 3. Results

Using the AERONET-OC data, the "blue" color indices Rrs(412)/Rrs(443) and Rrs(400)/Rrs(443) were calculated. Hereinafter, the color indices calculated on the basis of AERONET-OC data will be denoted as $CI^{AER}$(412/443) and $CI^{AER}$(400/443). Thus, the value of $CI^{AER}$(412/443) was found to be 0.77 ± 0.11, while $CI^{AER}$(400/443) was 0.72 ± 0.13. Based on this, the analysis of the statistical mode for the extended array of in situ data was carried out, and the value of the mode was equal to 0.8, which coincides with the average value obtained in the previous study for level 1.5 data [35] (Figure 3).

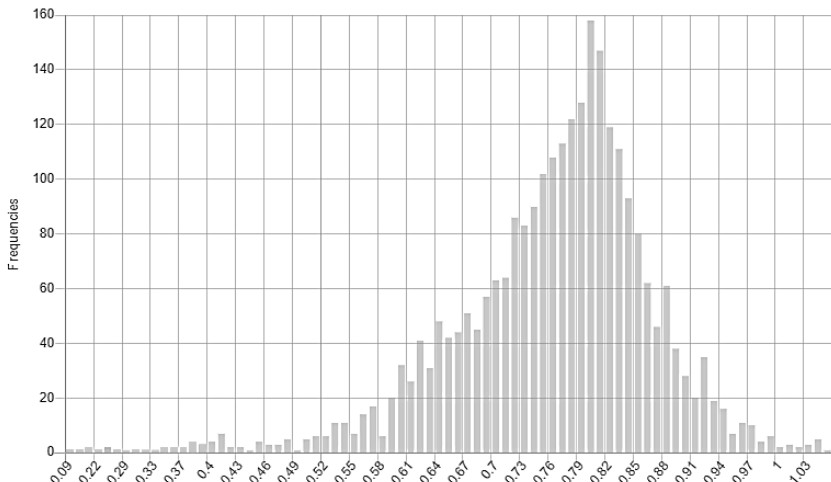

**Figure 3.** Frequency histogram for $CI^{AER}$(412/443) color index obtained from AERONET data during 2010–2022.

Similarly, $CI^{MHI}$(412/443) and $CI^{MHI}$(400/443) were calculated from 686 field measurements of Rrs during the MHI RAS cruises using the method described in Section 2.1.1, $CI^{MHI}$(412/443) = 0.836 ± 0.078 and $CI^{MHI}$(400/443) = 0.802 ± 0.098. Next, the linear regression $y = b \times x$ between Rrs(412) and Rrs(443) value and the possible measurement error were calculated. This procedure allows for minimizing the influence of errors in color index calculations performed at low values of the remote sensing reflectance. In this case, in the presence of linear dependence, the regression coefficient is the color index. As mentioned before, the measure of the uncertainty of using the color index in atmospheric correction procedures will be the value $SD(\lambda_1/\lambda_2)/(\lambda_2 - \lambda_1)$ (Table 2). To analyze the systematics of uncertainty errors, a graph of the dependence of the "blue" color index on the Rrs(412) value was additionally plotted (Figure 4).

**Table 2.** Average values of the "blue" color index from all field measurements in the Black Sea.

| Amount of Measurements | Bands | Average ± SD | Regression Coefficient, $b$ | $R^2$ | $SD/\Delta\lambda$ |
|---|---|---|---|---|---|
| **AERONET-OC** | | | | | |
| 1209 | 400/443 | 0.716 ± 0.127 | 0.723 | 0.957 | 0.0030 |
| 2620 | 412/443 | 0.77 ± 0.108 | 0.775 | 0.975 | 0.0035 |
| 1209 | 400/412 | 0.927 ± 0.096 | 0.930 | 0.990 | 0.0080 |
| **SPECTRUM-MHI** | | | | | |
| 686 | 400/443 | 0.802 ± 0.098 | 0.785 | 0.97 | 0.0023 |
| 686 | 412/443 | 0.836 ± 0.078 | 0.818 | 0.983 | 0.0025 |
| 686 | 400/412 | 0.958 ± 0.067 | 0.959 | 0.99 | 0.0056 |

It can be seen that the average values of $CI^{AER}(412/443)$ and $CI^{AER}(400/443)$ are smaller than $CI^{MHI}(412/443)$ and $CI^{MHI}(400/443)$ and have a larger spread (Table 2). Nevertheless, both indices fall into the adjacent area, taking into account the confidence intervals. Values of $CI^{AER}(400/412)$ and $CI^{MHI}(400/412)$ are close enough to each other, and their averages vary from 0.92–0.96, but it is worth noting that for CI(400/412), the values of $SD/\Delta\lambda$ more than twice as high in comparison to other. This fact contradicts the prediction made earlier (in the Introduction). Apparently, the increase in the error in the slope ($SD/\Delta\lambda$) is due to the opposite spectral dependences $L_w$ and $L_{sky}$ in the blue region, which enhances the influence of the uncertainty in estimating the reflected component of the skylight. Therefore, the use of CI(400/412) is unreliable and will not be considered further. According to the AERONET-OC data, a tendency for the "blue" color index to decrease toward low reflectance was observed (Figure 4). The array of Rrs(443) values has been sorted in ascending order by Rrs(412). In the figure, a moving average of the color index (over 21 points) is plotted depending on the average value of the sorted Rrs(412). It can be explained by an increase in the contribution of colored dissolved organic matter. However, it is worth noting that at high concentrations of CDOM, and even with neutral backscattering, the theoretical minimum index is 0.59 (the derivation of this value is discussed later in the Section 4), where a backscattering spectral slope equals 0.3, and absorption spectral slope equals to 0.018 nm$^{-1}$. Therefore, there is a systematic underestimation of the spectral radiance measured by AERONET-OC, while cruise measurements do not show this trend.

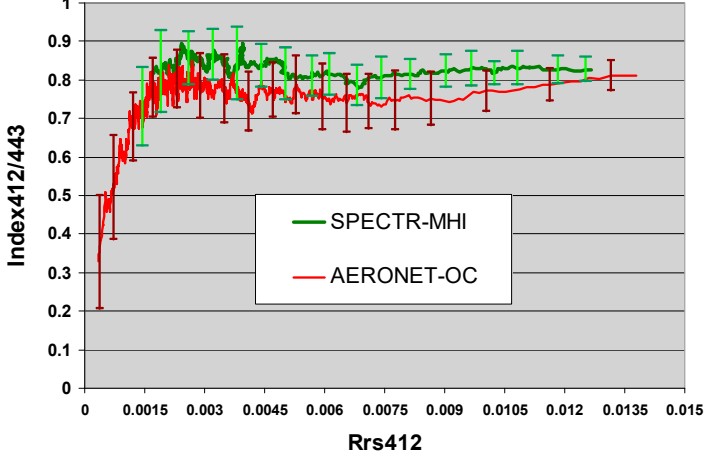

**Figure 4.** Dependence of $CI^{AER}(412/443)$ from Rrs(412).

Next, the variability of $CI^{sat}(412/443)$ was studied according to the remote sensing data. As previously noted, satellite measurements were used at the locations of AERONET-OC and SOP stations in the Black Sea. Later, for all the studied cases, the average $CI^{AER}(412/443)$ was calculated according to field measurements of AERONET-OC (Table 3).

It follows that the average $CI^{sat}(412/443)$ values from the SeaBASS data are underestimated, especially in the case of VIIRS SNPP measurements (Table 3). MODIS Aqua data are the closest to in situ measurements; however, in the case of using standard methods for processing remote sensing data, the standard deviation significantly increases.

**Table 3.** Average values of the "blue" color index from satellite measurements contained in the SeaBASS database for the coordinates of AERONET-OC stations in the Black Sea.

| Source | Amount of Measurements | $CI^{sat}(412/443)$ | $CI^{AER}(412/443)$ |
|---|---|---|---|
| MODIS Aqua | 650 | $0.76 \pm 0.20$ | $0.79 \pm 0.12$ |
| MODIS Terra | 821 | $0.69 \pm 0.24$ | $0.79 \pm 0.11$ |
| VIIRS SNPP | 669 | $0.67 \pm 0.18$ | $0.79 \pm 0.11$ |
| Sentinel OLCI 3A | 235 | $0.73 \pm 0.17$ | $0.79 \pm 0.11$ |

As mentioned before, the SOP in situ data were compared with OLCI data. Synchronous pairs of measurements from the SOP platform and OLCI data (Sentibel-3A and Sentinel-3B) from 2016–2021 were studied. For the analyzed periods, the average values of "blue" color indices were calculated from the field (SOP) and satellite values (OLCI) (Table 4). All satellite data with negative Rrs values in the range 400–443 nm were excluded from the analysis due to the presence of obvious atmospheric correction errors. For example, all synchronous satellite measurements for 2021 contained negative values at 400 nm, which explains the absence of corresponding color index values (Table 4).

**Table 4.** Average values of the "blue" color index from OLCI satellite measurements for SOP station in the Black Sea.

| Period | $CI^{MHI}(412/443)$ | $CI^{MHI}(400/443)$ | $CI^{sat}(412/443)$ | $CI^{sat}(400/443)$ |
|---|---|---|---|---|
| September 2016 | $0.91 \pm 0.08$ | $0.89 \pm 0.09$ | $0.94 \pm 0.24$ | $0.88 \pm 0.31$ |
| May 2017 | $0.89 \pm 0.02$ | $0.87 \pm 0.02$ | $0.80 \pm 0.03$ | $0.62 \pm 0.03$ |
| October 2017 | $0.86 \pm 0.04$ | $0.85 \pm 0.04$ | $0.76 \pm 0.05$ | $0.61 \pm 0.22$ |
| October 2018 | $0.83 \pm 0.09$ | $0.79 \pm 0.1$ | $0.58 \pm 0.16$ | $0.41 \pm 0.23$ |
| June 2019 | $0.81 \pm 0.05$ | $0.76 \pm 0.06$ | $0.72 \pm 0.006$ | $0.60 \pm 0.014$ |
| July 2021 | $0.82 \pm 0.07$ | $0.81 \pm 0.1$ | | |
| **Average** | **$0.85 \pm 0.06$** | **$0.82 \pm 0.07$** | **$0.76 \pm 0.09$** | **$0.62 \pm 0.15$** |

The table highlighted the time interval (September 2016) when increased values of color indices were observed, which corresponded to more transparent waters than usual. For example, on 25 September 2016, the values of Secchi depth = 22 m were measured.

Based on Table 4, it can be seen that the OLCI index values, although they fall within the confidence intervals, have a large dispersion. This is especially pronounced for the calculation of the color index CI(400/443) since large uncertainties are observed at the 400 nm band associated with incorrect accounting for the atmospheric component. These results once again substantiate the need for reference to in situ measurements of the "blue" color index.

## 4. Discussion

When changing the type of water from Case 1 to Case 2, the color index decreases to values <1. In the Black Sea, Rrs($\lambda$) is monotonously increasing to its maximum around 490 nm, which means that the "blue" color index is strictly less than 1. During the review of similar studies, this fact was confirmed for various water areas: the Mediterranean Sea, the Red Sea, and the Beaufort Sea [27,30,31]. Based on a limited set of in situ measurements, it was also revealed that CI(412/443) = 0.8 for the waters of the Black Sea northwestern part [34]. We have made a theoretical assessment of the limits of color index changing, which depend on the spectral properties of backscattering and absorption by optically significant materials. The minimum color index can be observed in very turbid waters,

provided that the backscattering is formed mainly by large particles of biological origin. On the contrary, due to a decrease in the contribution of light scattering on particles in the open part of the Black Sea, the color index will be maximal.

Let $n$ be the exponent obtained by approximating the total backscattering $b_b$, which is the sum of particulate backscattering $b_{bp}$ and water molecules backscattering $b_{bw}$, by the power law in the range 412–443 nm. The value of $n$ is estimated as $n \approx \frac{\ln[b_b(412)/b_b(443)]}{\ln(443/412)}$.

Spectral properties of light absorption in water depend on the concentration of optically significant materials. According to the Black Sea Atlas of Mankovsky et al. [60], the light absorption of CDOM predominates in "blue", where the approximate contribution of detritus, phytoplankton, and CDOM into total absorption is estimated as 1:1:8. We approximated light absorption of water in "blue" by the law $a(\lambda) \approx \exp[\gamma(400 - \lambda)]$, where $\gamma$ was the absorption spectral slope varying from 0.008 up to 0.018 nm$^{-1}$.

According to Rrs $\sim b_b/a$ the "blue" color index was calculated as

$$CI\left(\frac{412}{443}\right) = \frac{b_b(412) \cdot a(443)}{b_b(443) \cdot a(412)} \tag{5}$$

The maximum value of the total backscattering wavelength exponent was taken equal to 3. If we take the maximum slope as $b_{bp}(\lambda) \sim \lambda^{-1.7}$ (see [61]) and use Morel's formula for backscattering of pure seawater [62]:

$$b_{bw}(\lambda) = 0.002913 \cdot (1 + S/37) \cdot [400/\lambda]^{4.32}, \tag{6}$$

where $S$ is the salinity in ‰ ($S$ = 18‰ for the Black Sea), then the value $n = 3$ would be $b_{bp}(443) = 0.0024$ m$^{-1}$, which is even less than the background value in the green area for the Black Sea $b_{bp}(555) = 0.0025$ m$^{-1}$ [63].

The simulation results are presented in Table 5.

**Table 5.** Color index CI(412/443) as function of the backscattering and absorption spectral characteristics.

| Total Backscattering Wavelength Exponent | 0.3 | 0.6 | 0.9 | 1.2 | 1.5 | 1.8 | 2.1 | 2.4 | 2.7 | 3.0 |
|---|---|---|---|---|---|---|---|---|---|---|
| **Absorption Spectral Slope, nm$^{-1}$** | | | | | $CI\left(\frac{412}{443}\right)$ | | | | | |
| 0.008 | 0.798 | 0.815 | 0.833 | 0.851 | 0.87 | 0.889 | 0.909 | 0.929 | 0.949 | 0.970 |
| 0.010 | 0.75 | 0.766 | 0.783 | 0.8 | 0.818 | 0.836 | 0.854 | 0.873 | 0.892 | 0.912 |
| 0.012 | 0.705 | 0.72 | 0.736 | 0.752 | 0.769 | 0.786 | 0.803 | 0.82 | 0.839 | 0.857 |
| 0.014 | 0.662 | 0.677 | 0.692 | 0.707 | 0.722 | 0.738 | 0.755 | 0.771 | 0.788 | 0.805 |
| 0.016 | 0.622 | 0.636 | 0.65 | 0.664 | 0.679 | 0.694 | 0.709 | 0.725 | 0.741 | 0.757 |
| 0.018 | 0.585 | 0.598 | 0.611 | 0.624 | 0.638 | 0.652 | 0.667 | 0.681 | 0.696 | 0.712 |

The calculated model values of CI(412/443) confirm that even at maximum absorption, the values of CI(412/443) cannot be lower than 0.585. This is indirectly supported by the previously obtained results presented in [30,31]. This information can also be used to "filter" remote sensing data. During the study, it was shown that the satellite values of the color index have a large scatter. Based on the data of Table 2, it was calculated that for MODIS Aqua data, 15% of data are physically incorrect according to the selection criterion CI(412/443) < 0.59, 30% for MODIS Terra, 20% for Sentinel 3A, 26% for VIIRS SNPP. In general, based on Table 4, it can be seen that the limits of change for CI(412/443) are in the range from 0.59 to 0.97, which is consistent with the results obtained during the study. The results are in good agreement with [7], where all data from the SeaBASS validation database were used as input data, and the CI(412/443) ratios (in the case of VIIRS CI(410/443)) were calculated. It was found that these ratios for in situ measurements (from look-up tables) ranged from 0.6 to 1.4, while the satellite products had a much larger scatter, namely, for the SeaWiFS data, the ratios varied from 0.2 to 1.6, and from VIIRS and MODIS Aqua data, CI(410/443)) was from 0.1 to 1.1. It should be noted that the highest

concentration of all studied ratios was within the range of 0.6 to 1.0. Therefore, this work is another example of the expediency of using the condition CI(412/443) > 0.59 as a certain criterion for atmospheric correction error. In addition, according to Table 4 values, the color index should strongly depend on the ratio of absorption by suspended particles to CDOM absorption. To a lesser extent, there is a dependence on the spectral slope of backscattering.

When analyzing an extended set of field measurements, it was proved that the "blue" color index varies very slightly in the Black Sea. This statement is substantiated by the use of independent sources of measurements of the spectral sea reflectance and a long-term array of observations. Regardless of seasonal phenomena, for example, the presence of phytoplankton blooms, the concentration of dissolved substances in water, atmospheric conditions, the type of water (coastal or open), the average color index in the short-wave region CI(412/443) for the Black Sea varies from 0.77 to 0.83. The most observed value was 0.8. During the study, it was found that even under conditions with increased content of coccolithophores in the upper water layer, the average $CI^{MHI}(412/443)$ ratio was $0.801 \pm 0.003$. The in situ CI(400/443) values have a more noticeable variability from 0.716 to 0.80, and since the 400 nm band is not so widely used by satellite measurement processing algorithms, it has been possible to test it using OLCI Santinel-3 data only.

As mentioned earlier in the work [34], an algorithm for additional correction of Ocean Color level 2 satellite data was developed to minimize atmospheric correction errors during dust transfers (see details in the Section 1). The efficiency of this algorithm was tested on the basis of field measurements from AERONET-OC stations in the Black Sea. In this paper, we tested this algorithm based on MHI RAS in situ data. This check is another example of the possibility of using "blue" color indices as a reference for remote sensing of Black Sea water. We used independent measurements of the remote sensing reflectance obtained during the cruise of the R/V "Professor Vodyanitsky" (April–May 2021). For each measurement, synchronous (obtained on the same day) Rrs($\lambda$) product was found for the MODIS scanner from the Aqua and Terra satellites. Data from the last reprocessing (R2022) were used. After applying an additional model correction for 90 possible measurements, the Rrs($\lambda$) spectra become closer to the actually measured spectrum, which indicates that the method for taking into account the errors of the standard atmospheric correction works successfully. For a detailed illustration of the operation of the error correction procedure, two synchronous pairs for April 3, 2021 and April 30, 2021 were selected.

The following examples illustrate the operation of an additional correction algorithm based on the interpolation of the estimated atmospheric correction error from the short-wave and near-IR regions to the mid-visible range (Figure 5). Let us assume that the error of the standard algorithm is function $f(\lambda)$, and $f(870) = 0$. Then, the proportionality coefficient is found from the equation:

$$\text{Rrs}(412) + C \cdot f(412) = CI\left(\frac{412}{443}\right) \cdot \text{Rrs}(443) + C \cdot CI\left(\frac{412}{443}\right) \cdot f(443) \qquad (7)$$

When substituting the blue index as CI(412/443) = 0.8 and $f(\lambda) \sim \lambda^{-4} - 870^{-4}$ the new range of Rrs values will be calculated by the formula:

$$\text{Rrs}^*(\lambda) = \text{Rrs}(\lambda) + \frac{0.8 \cdot \text{Rrs}(443) - \text{Rrs}(412)}{(\lambda/412)^4 - 0.8 \cdot (\lambda/443)^4 - 0.2 \cdot (\lambda/870)^4}\left[1 - (\lambda/870)^4\right] \qquad (8)$$

As a result of additional correction, the values become much closer to the in situ data. In the first case (3 April), the Rrs spectra were very different, and the color index was 0.53. For the data for 30 April, the differences between satellite and field Rrs are not so significant (color index 0.67). Nevertheless, in both cases, the corrected spectra corresponded better to field measurements and turned out to be fairly close to each other. The additional correction algorithm uses an exponent of $-4$, which makes the interpolation function quite steep, and the correction in the red is negligible. For this reason, and also for simplicity, the condition was not previously applied in [34].

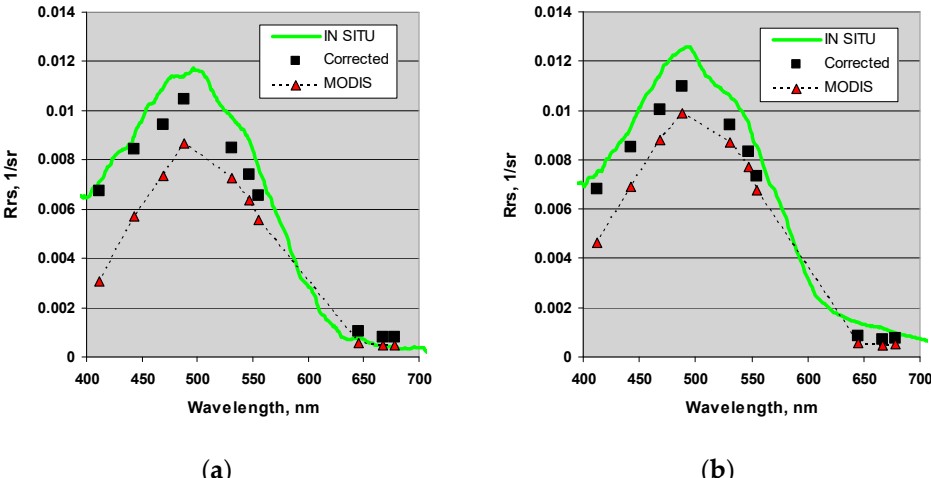

**Figure 5.** Comparison of field Rrs(λ) of the Black Sea with MODIS Aqua satellite values before and after correction, 2021, 3 April (**a**) and 30 April (**b**).

Another point of this study is that it demonstrated the need to improve the accuracy of field measurements of the water-leaving radiance, which is further converted into remote sensing reflectance. It is shown that in situ measurements from the AERONET-OC platforms and MHI RAS data have a number of systematic errors and uncertainties. For AERONET-OC data, these uncertainties are associated with the neglect of polarization effects, the influence of which determines the dependence of the reflection coefficient on the wavelength. In the case of applying the methodology of the MHI RAS data, the errors are associated with a possible difference in the roughness of waves in the dark absorbing cuvette and the sea. This fact should be taken into account because the quality of validation of satellite products strongly depends on the quality of field measurements.

Comparative analysis of satellite and field measurements was carried out for three stations: two AERONET-OC stations (Gloria, Galata) and SOP. The values of the color index CI(412/443) were analyzed using satellite data from scanners: MODIS Aqua/Terra, VIIRS SNPP, and OLCI Sentinel 3A. All satellite data showed a large scatter based on the size of the standard deviation of the sample (Tables 3 and 4). For AERONET-OC stations, the most reliable result was obtained for the MODIS Aqua data, where CI(412/443) = 0.76. However, when analyzing data from the SOP station, OLCI also demonstrated good comparability of calculations. However, large uncertainties were found, especially for the calculation of the color index CI(400/443), since large uncertainties are observed on the 400 nm band associated with incorrect accounting for the atmospheric component. These results once again substantiate the need for reference to in situ measurements of the "blue" color index.

When using this information to "filter" remote sensing data, it was demonstrated that for MODIS Aqua, 15% of data are physically incorrect according to this selection criterion, 30% for MODIS Terra, 20% for Sentinel 3A, 26% for VIIRS SNPP. In the course of the work, it was shown that the MODIS Aqua satellite has fewer outliers in the data; this can be explained by several reasons: a larger number of spectral channels (compared with VIIRS SNPP), a long commissioning period (a large number of reprocessing to improve the quality of satellite data). For example, OLCI is relatively new and has only the first reprocessing with the prospect of improved versions.

In the future, it is planned to calculate similar combinations of "blue" color indices and apply the proposed algorithm for retrieving the remote sensing reflectance for other water areas. First, the use of this method will be tested for areas with a large contribution of CDOM and confirmed dust transport, such as the Yellow and East China Seas.

## 5. Conclusions

The main conclusion of this work is that the "blue" color index can be used as a reference value in atmospheric correction algorithms. It was found that the optimal variation of

CI(412/443) for the Black Sea is the values within the range from 0.77 to 0.83. The model values of CI(412/443) were also calculated, which confirms that even at maximum absorption, the values of CI(412/443) cannot be lower than 0.585. Analysis of in situ reflectance data showed that the color index Rrs(412)/Rrs(443) has much lower variability compared to Rrs itself. It was found that for the Northern part of the Black Sea, this color index varies from 0.77 to 0.83. The value of Rrs(412)/Rrs(443) = 0.8 was applied as a reference to correct the spectra of the remote-sensed reflectance in addition to the standard atmospheric correction.

The lower limit of the color index Rrs(412)/Rrs(443) was established theoretically. It was confirmed that even in the case of large concentrations of light-absorbing constituents in the seawater, this index cannot be lower than 0.585. This fact allows filtering remote-sensed data not only on the basis of positive/negative values at 400–443 nm and error flags but to automatically check the correct retrieval of reflectance spectral shape.

**Author Contributions:** Conceptualization, E.S. and A.P.; methodology, E.S. and A.P.; software, E.S., A.P., V.S. and E.K.; validation, A.P. and V.S.; formal analysis, E.S. and V.S.; investigation, E.S., A.P., V.S. and E.K.; resources, A.P. and E.K.; data curation, A.P. and E.K.; writing—original draft preparation, A.P. and E.K.; writing—review and editing, E.S. and V.S.; visualization E.S., A.P., V.S. and E.K.; supervision, E.S. and V.S. All authors have read and agreed to the published version of the manuscript.

**Funding:** The work was carried out within the framework of the state task theme FNNN-2021–0003 "Development of operational oceanology methods based on interdisciplinary research of processes of the marine environment formation and evolution and on mathematical modeling using data of remote and contact measurements" ("Operational oceanology").

**Data Availability Statement:** The data presented in this study are available on request from the corresponding author. Upon request, we are ready to provide a measurement protocol and device characteristics of the MHI spectrophotometer.

**Acknowledgments:** The authors thank Giuseppe Zibordi for processing of measurements obtained at the Galata_Platform and Gloria AERONET stations and for the possibility of using high-quality in situ ocean color measurements; NASA Goddard Space Flight Center, Ocean Ecology Laboratory, Ocean Biology Processing Group; (2018) for the freely distributed remote sensing information in OceanColor website.

**Conflicts of Interest:** The authors declare no conflict of interest.

## Appendix A

Here, we give some information to prove the choice of measuring using a cuvette.

The main cause for using the cuvette was to take into account polarization effects. Let $\rho_{sky}(\lambda)$ be the ratio of the radiance of the sky, at the point from which the light enters into the photometer, to the irradiance at sea level. Then, the reflectance of the water body will be equal to

$$\rho_w(\lambda) = \rho_s(\lambda) - Rf(\lambda) \cdot \rho_{sky}(\lambda) \tag{A1}$$

where $Rf$ is the effective Fresnel's coefficient, depending on both the sea roughness and polarization parameters of the incident light, such as polarization degree and the location of the polarization plane. Polarization degree is a function of wavelength and therefore $Rf$ is too. For this cause, it is not sufficient to measure just the radiance of the sky. Since the cuvette is filled with water, the reflection from its surface is similar to the reflection from the sea surface.

The cuvette makes it possible to measure the reflected sky radiance in the same observation geometry as the upwelling radiance.

We consider the simplest case when the sea surface is smooth, and the light is scattered only on the molecules. Fully polarized light is reflected according to the formula:

$$Rf_{eff}(\theta) = Rf_{\perp}(\theta) \cdot \cos^2 \alpha + Rf_{II}(\theta) \cdot \sin^2 \alpha = Rf(\theta) \cdot \left(1 + P_f \cdot \cos 2\alpha\right) \tag{A2}$$

$$Rf(\theta) = \frac{1}{2}(Rf_{II}(\theta) + Rf_{\perp}(\theta)), \tag{A3}$$

$$P_f = \frac{Rf_\perp(\theta) - Rf_{II}(\theta)}{Rf_\perp(\theta) + Rf_{II}(\theta)}. \tag{A4}$$

where $Rf_{eff}(\theta)$ is the efficient Fresnel coefficient as a function of the incident zenith angle $\theta$;
$Rf_\perp(\theta)$, $Rf_{II}(\theta)$ are Fresnel coefficients for s-polarized light and for p-polarized light;
$Rf(\theta)$ is the Fresnel coefficient for unpolarized light;
$P_f$ is the degree of polarization due to Fresnel reflection;
$\alpha$ is the angle between the plane of polarization and the plane of reflection.

Let $P_a$ be a degree of polarization for the light scattered in the atmosphere (incident light). Then, the reflection coefficient will be equal:

$$Rf_{eff}(\theta, \lambda) = Rf(\theta) \cdot \left(1 - P_a(\lambda, \gamma) + P_a(\lambda) \cdot \left[1 + P_f \cdot \cos 2\alpha\right]\right) = \\ Rf(\theta) \cdot \left(1 + P_a(\lambda, \gamma) \cdot P_f \cdot \cos 2\alpha\right) \tag{A5}$$

where $P_a$ is wavelength-dependent. The value of $\cos 2\alpha$ can be obtained from the formula for a spherical triangle whose vertices are: Z—zenith; S—the Sun; M—the Mirror point (a point in the sky corresponding to the direction of observation). The sides of the triangle are ZS—Sun zenith angle $\theta_s$; ZM—zenith angle of M point $\theta$; MS—the scattering angle $\gamma$. If $\phi$ is the Sun azimuth minus observation azimuth, then

$$\sin \alpha = \frac{\sin \theta_s}{\sin \gamma} \sin \phi \tag{A6}$$

where $\cos \gamma = \cos \theta \cos \theta_S + \sin \theta \sin \theta_S \cos \phi$.

The next figure illustrates the influence of polarized light in the atmosphere on Fresnel reflection for some fixed observation angles. The polarization degree of the atmosphere was taken as

$$P_a(\gamma) = \frac{1 - \cos^2 \gamma}{1 + \cos^2 \gamma} \tag{A7}$$

Figure A1 shows that the uncertainty can reach tens of percent. For azimuth 90° at solar zenith angles of about 32°, the effect of polarization on the reflection coefficient is minimal. Due to polarization, the reflection coefficient decreases monotonously with the Sun zenith angle; for instance, at $\theta_S = 45°$, the resulting reflection coefficient becomes 17% lower. Under real atmospheric conditions, the light scattering by aerosol decreases the polarization degree of scattered light. The ratio of aerosol scattering to molecular scattering grows with wavelength. So, the consideration of polarization could not be so critical in longer wavelengths. Unfortunately, there is no criterion of quality for radiance retrieval in the short wavelengths, while in the NIR, the sea reflectance should be zero. However, one conclusion that the reflected sky component, as a rule, is overestimated can be drawn.

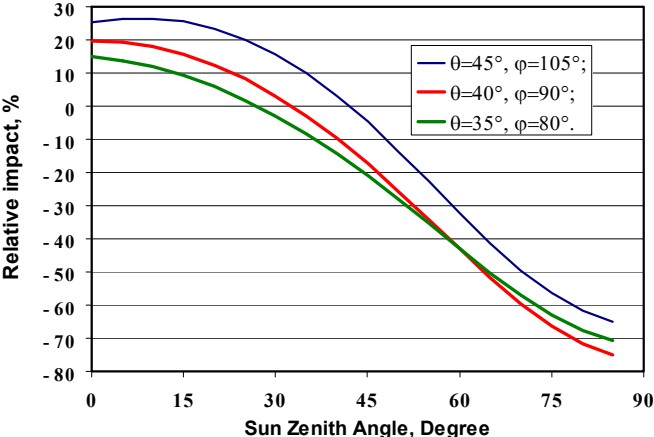

**Figure A1.** Relative impact of polarization effects. Values $P_a \cdot P_f \cdot \cos 2\alpha$ are plotted versus the Sun's zenith angle.

This example shows the need to take into account the polarization effects during the sea reflectance calculation. In the MHI RAS measurement method, this consideration is carried out experimentally. As the practice of using the dark cuvette method has shown, due to the high accuracy of the measured spectra of the water-leaving radiance, it became possible to determine the concentrations of optically significant materials in seawater with acceptable accuracy [41].

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
