# Peer review of "Blue Color Indices as a Reference for Remote Sensing of Black Sea Water"

_remotesensing, doi:10.3390/rs15143658_

Round 1

Reviewer 1 Report (New Reviewer)

Dear authors,

I found your paper of large interest to the community as it suggest a way to help constrain atmospheric correction using field data. The finding that a reflectance ratio is well constrained is exciting and could make the Black Sea a good test case for evaluation of ocean color remote sensing.

My major issues are with:

1. The paper is poorly organized with materials that should be in the 'method' section in the 'results' section and material that should be in the 'discussion' section in the 'results' section.

2. There are too many typos and poor stylistic choices such as starting a sentence with 'figure 2 shows....'. Tell the reader what you observed and reference the figure or table in parenthesis at the end of the sentence.

3. The argument supporting the 0.8 reflectance ratio could be significantly streamlined. The equation in L. 594 makes this ratio to a large degree amplitude independent such that table 5 is the main explanation to the constrained values observed. Fixing the values of beam attenuation and absorption is not necessary and is likely wrong as IOPs are highly variable in the environment (if you insist on fixing them show us time series where they are indeed very constrained over the whole year).

This paper will be of significantly more value if it were easy to read. Currently it has too many typos and the presentation can be significantly improved.

Author Response

Dear Reviewer,

Thank you for your valuable comments ans suggestions. Our answer is in the atteched file.

Reviewer 2 Report (New Reviewer)

Manuscript Number: RS-2454813

Title: Blue color indices as a reference for remote sensing of Black Sea water

Authors: Evgeny Shybanov, Anna Papkova, Elena Korchemkina and Vyacheslav Suslin

1. Recommendation:

Major revision

2. Overview and general recommendation:

The manuscript by Evgeny et al. presents a study of using blue color indices as a reference for remote sensing of Black Sea water based on field data from 2002-2021. The current topic is of interest which can present an automatically check for the correct retrieval of reflectance spectral shape. The article is meaningful and its layout is neat and appropriate. However, certain discussions and conclusions still require the inclusion of relevant supporting charts. Furthermore, the discussions lack sufficient citation of relevant references. It is recommended to supplement the article with the necessary evidence in the form of charts and graphs, and to incorporate relevant references to support the discussions. The decision of this review is ‘Major revision’. Some major and minor comments were listed as follows.

3. Major comments:

(1) It is suggested to incorporate the highlights mentioned in the conclusion into the abstract, such as which satellite sensor’s blue index is closest to the measured value, and to explain the possible reasons as much as possible.

(2) Too many key words. It is suggested to be no more than 6 key words.

(3) This article lacks an accompanying image or verification results that demonstrate the role of the blue index in spectral correction.

(4) Please incorporate relevant references to support the discussions.

4. Minor comments:

(1) Line 48-49: Please merge this sentence with the previous paragraph.

(2) Line 79-80: please revised “it is more correct to use the band ratio this is due to backscattering changes including coccolithophore blooms, therefore, the use of the difference can lead to large errors” into “The band ratio is recommended as backscattering changes including coccolithophore blooms can lead to large errors when using band difference”. Please add reference to this sentence too.

(3) Line 112: “In [30], it was found that the average…” need to be revised as “Lee and Hu documented that the average…[30]”. It is recommended to modify the writing style of “in” followed by a reference number in the entire text.

(4) The expressions similar to “The authors believe” throughout the entire text are recommended to be revised into the passive voice. For example in Line 116 The authors believe that this is a clear sign of excessive CDOM absorption” can be revised into “This is considered a clear sign of excessive CDOM absorption”.

(5) Line 185: “we applya larger set” should be “A larger set of … were applied…”.

(6) Line 449-455: Is there an entire paragraph missing with only the single word “From”?

(7) The subsequent part of the article is missing formula numbers throughout from Line 582.

(8) Line 675: “4. Conclusion” should be “5. Conclusion”.

Author Response

Dear Reviewer,

Thank you for your valuable comments ans suggestions. Our answer is in the atteched file.

Round 2

Reviewer 2 Report (New Reviewer)

Manuscript Number: RS-2454813-round2

Title: Blue color indices as a reference for remote sensing of Black Sea water

Authors: Evgeny Shybanov, Anna Papkova, Elena Korchemkina and Vyacheslav Suslin

 1. Recommendation:

Can be accepted for publication after revising follow minor comments.

 2. Minor comments:

1)        Line 2: “Black sea” in the title should be “Black Sea”.

2)        Line 19: “datashowed” should be “data showed”.

3)        Line 110: “seasonal variability In this study” should be “seasonal variability. In this study”.

4)        Line 154: Equation 2 is an image, please re-edit.

5)        Line 256: Please standardize the “Rrs” text formatting throughout the text, either all italics or all block font.

6)        Line 475-489: There are some formatting errors, as follows

7)        Line 575 and 580: The formatting of “Rrs~bb/a” and “bbp(λ)~λ-1.7” are not the same as full-text formatting.

8)        Line 635-636: “during cruise 116 of the R/V 635 Prof. Vodyanitsky" from April to May 2021.” is confused, please revise.

Author Response

 Dear Reviewer, Thank you for your help. We have made the required corrections, they are highlighted yellow in the text.

This manuscript is a resubmission of an earlier submission. The following is a list of the peer review reports and author responses from that submission.

Round 1

Reviewer 1 Report

This manuscript presents blue color indices for the Black Sea and demonstrates their utility in improving atmospheric correction errors of remote sensing reflectance in the blue portion of the spectrum.

I found the introduction hard to follow: the motivation for this study was not clearly articulated, and I had to jump into a lot of the cited references to fully understand what the authors were saying. The connection between the blue color indices and atmospheric correction are not well stated. Throughout the text there are places where there are statements made that are not clear and it would be beneficial to have some more details provided in the text (see specific comments section below).

My biggest concern with this paper was the in situ MHI RAS reflectance measurement approach. See detailed comments below.

My initial thought was that the MHI RAS part of the paper should be removed and focus should just be placed on the AERONET-OC and satellite components. But it seems a lot of the AERONET-OC analysis was done in a previous paper (ref [29]), which leaves the question of what new aspect would be left?

Reflectance measurements:

- Is R_p the bidirectional reflectance function (BRDF)?

- What does “cross-calibration with Spectralon reflectance standard” mean? Did you use the reflectance coefficient provided with the panel? Did you measure it yourself? Can you please provide more details of this method?

- Remote sensing reflectance is the water leaving radiance (Lw) divided by the downwelling irradiance (Ed). In line 205 the authors state they are measuring downwelling radiance NOT irradiance. If the authors are indeed estimating downwelling radiance (Ld) using the panel, then the final reflectance they calculate is not the remote sensing reflectance, but the radiance reflectance (= Lw / Ld). However, if the R_p is the BRDF, then pi*Lp/Rp is actually the downwelling irradiance, and the authors are indeed calculating Rrs. Can you please clarify which you are measuring and update the text accordingly? i.e. are you really estimating the downwelling irradiance OR are you calculating the radiance reflectance?

- In line 191 the authors state they are measuring “the total upwelling radiance Lu above the water surface (fig 2a)”. The arrows in figure 2a shows they are measuring the total water leaving radiance and the sky reflected radiance (Lr), which is indeed the total upwelling radiance. But in Fig 2a, Lu is labeling the water leaving radiance i.e. Lu is NOT the total upwelling radiance measured by the notation shown in Fig 2a, it is the water leaving radiance only, thus the total upwelling radiance measured is Lu + Lr using the notation in Fig 2a. To be consistent in notation used in Figure 2 and with the language in the text, both here and in Equation (3), a different symbol should be used in Fig 2a (e.g. Lw in place of Lu, which is commonly used within the literature, and is the notation used in the protocol referenced).

- To calculate the water leaving radiance requires measurement of upwelling surface radiance (Lu) i.e. a combination of the water leaving radiance (Lw) and the sky radiance (Lsky). The sky radiance is used to calculate the sky reflected radiance and subtracted from the total upwelling surface radiance to calculate the water leaving radiance: Lw = Lu - rho*Lsky, where rho = sea surface reflectance factor. The protocol for using a reflectance panel/plaque for measuring remote sensing reflectance still requires a measurement of Lsky (p22 and Eq 3.3 of the protocol referenced by the authors). With the setup described by the authors, they are not making any measurement of Lsky. The authors state they are measuring the sky reflected radiance using the setup shown in Fig2b i.e. by-passing the need for measuring Lsky. But there will still be some water leaving signal contributing to the signal measured in the absorbing cuvette setup – or if not, I’d like to see some testing or evidence showing this signal is negligible. Furthermore, a big factor in the amount of sky radiance that is reflected off of the water’s surface is driven by conditions of the sea surface i.e. the amount of light reflected is different for a rough surface vs a calm surface. By measuring the reflected radiance in the absorbing cuvette, the surface conditions are very different from the natural environment, thus the measured reflected radiance is not representative of the sky radiance reflected from the sea surface. Consequently, I don’t think the described setup is accurately measuring remote sensing reflectance.

Specific Comments:

L30: CDOM = colored dissolved organic matter NOT dissolved matter. CDOM is a subset of the whole dissolved matter pool.

L31: what is meant by “the ratio of impurity concentrations”? What does “impurity” refer to? Non-phytoplankton? Or any optically significant material? Can this please be clarified in the text?

L69-75: blue color index introduced.

“The following variations are used most often” – please provide references.

The authors state band ratios. Normally, color indices are band differences, not band ratios e.g. see Hu et al. (2012), Le et al. (2018), Mitchell et al. (2017).

Hu, Chuanmin, Zhongping Lee, and Bryan Franz. “Chlorophyll a Algorithms for Oligotrophic Oceans: A Novel Approach Based on Three-Band Reflectance Difference.” Journal of Geophysical Research: Oceans 117, no. 1 (2012): 1–25. https://doi.org/10.1029/2011JC007395.

Le, Chengfeng, Xueying Zhou, Chuanmin Hu, Zhongping Lee, Lin Li, and Dariusz Stramski. “A Color-Index-Based Empirical Algorithm for Determining Particulate Organic Carbon Concentration in the Ocean From Satellite Observations.” Journal of Geophysical Research: Oceans 123, no. 10 (2018): 7407–19. https://doi.org/10.1029/2018JC014014.

Mitchell, C., C. Hu, B. Bowler, D. Drapeau, and W. M. Balch. “Estimating Particulate Inorganic Carbon Concentrations of the Global Ocean From Ocean Color Measurements Using a Reflectance Difference Approach.” Journal of Geophysical Research: Oceans 122 (2017): 8707–20. https://doi.org/10.1002/2017JC013146.

L101-102: “Therefore, this ratio can be effectively used to estimate the contribution of the atmosphere”. Where is this conclusion coming from? The referenced paper is a compiled dataset of in situ bio-optical data for ocean-color validation. It does not discuss atmospheric signals.

L112: started using the notation CI(412/443) without explaining it

L154: Suggest word choice change - I thought it was shown from in situ data in [29] that the blue color index had low variability, not that it was “assumed” the blue color index had low variability.

L270: a couple of typos: urn should be um (i.e. micro meters) and the three 371m bands should be “I1-I3” not “11 – I3”.

L274: Please provide a reference for the statement that OLCI is the “most accurate remote sensing instrument”

L309: what is the difference between Rrs(412)/Rrs(443) and CI^AER(412/443)? Please define the CI^AER notation in the text.

L311, reference [29]: I checked this study, and it seems like the 0.8 +/- 0.8 value for CI^AER(412/443) referenced was for AERONET-OC data spanning 2011-2021 (Fig 3 in the ref [29]), not for before 2018 as stated here. In fact, seeing this makes me question what the purpose of this study is – it seems as if this study is only adding one more year of data to the results shown in [29].

L313: How does the mode of the data analyzed in this study matching with the mean of a previous study “explain the previously obtained result”?

L323: What “possible measurement errors” were calculated? What does regressing a variable against that variable divided by another value show us?

Fig 4 and Table 2: Are the regression coefficients and R^2 for 412/443 in Table 2 from the data shown in Fig 4? I find it hard to imagine that is the case – the R^2 values are too high for a data that don’t have a linear relationship, and the slope values (the regression coefficient, b) are too close to 1 for variables that have very different ranges of values. If these data are indeed different, can you please clarify what the difference is (from the way the text currently is written it seems like they should be the same)? If these data are indeed the same, please can you include a figure showing the regression fit to the data?

L346-349: backscattering spectral slope and absorption spectral slope

L361: “coccolithophore blooms is” not “coccolithophores bloom”

L397: why only use that specific cruise and not all the cruise data available?

Fig 5 (b): this figure feels a bit out of place. There’s no real discussion on AOT in the paper, and there’s no discussion about why this figure is included in the paper or how it relates to any of the paper’s argument.

L434-435: Please provide a reference for this statement – it’s the first time this is stated as such. In the introduction there’s discussion about the blue color index and Case 1 vs Case 2 waters, but there is no statement in there about this clear defining line of a blue color index below 1 being Case 2 and above being Case 1.

Fig 6: Because these lines are all means, can you please provide error bars on the spectra? All the satellite values are very close to each other and I wonder if they are statistically different from each other.

L448: “optically significant materials” not “impurities”

L454 & eqn 5: can more details about the relationship used for Rrs412 (Rrs412 = 0.006/pi) and the form of Eqn (5) please be provided? There’s a lot of literature regarding the proportionality of Rrs with bb/a and the proportionality constant to use – why did the authors use this particular choice of 0.15/pi? Furthermore, why is this particular value used for Rrs412? A different value choice would alter all the following calculation results.

Table 4: I’m not following the described simulations. The text from L446 – 481 walks through a series of assumptions and calculations that results in a value of n = 3.07 (the backscattering spectral slope exponent), so why does Table 4 have a range of n values use?

Author Response

Dear Reviewer!

Thank you for giving us the opportunity to submit a revised draft of the manuscript “Blue color indices as a reference for remote sensing of Black sea water” for publication in the Remote Sensing journal. We appreciate the time and effort that you dedicated to providing feedback on our manuscript and are grateful for the insightful comments on and valuable improvements to our paper. We have incorporated most of your the suggestions and recommendations. Those main changes are highlighted within the manuscript (yellow). Please see below, for a point-by-point response to your comments and concerns (green highlighted). All page numbers refer to the revised manuscript file with tracked changes. In addition, new sources of citation were added to the text of the manuscript, and therefore the previous reference numbering was changed.

Kind Regards!

Reviewer 2 Report

The manuscript is very relevant.

Despite technological progress in optical scanners, there is a problem with the accuracy of satellite algorithms. One of the reasons for such inaccuracy of the algorithms is the atmospheric correction.

Therefore, the main achievement of these researches related to the refinement of the atmospheric correction could successfully applied for the remote sensing assessment of the Black Sea ecosystem state.

The manuscript can be published after minor revision.

As the main comment I would notice the "regional" level of the research.

There are a few minor comments

Row 28: “ “For such water areas, the balance of nutrients is disturbed, and as a result statistical relationships between the concentrations of chlorophyll and dissolved matter (CDOM) are weakened, 30 while the ratio of impurity concentrations is strongly shifted towards nonliving organic 31 matter - detritus and CDOM.” The authors absolutely unreasonably write about the influence of nutrient availability on the formation of Case 2 waters (domination of CDM in light absorption).

Row 150, 356, 361, 363 etc: I recommend to replace “bloom” by “phytoplankton “bloom””

Row 28: The authors use in correctly “dissolved matter” for CDOM. Colored dissolved organic matter. It should be corrected in all cases.

Row 356: I recommend to replace “massive blooms” by “intensive phytoplankton “blooms””

Row 363: I recommend to replace “abnormal blooms” by “extra intensive “blooms””

Row 115:  CI (412/443) → CI(412/443)

Row 130: CI (410/443) → CI(410/443)

Row 141: k → k

Row 266: VIIRS-SNPP → VIIRS SNPP

Row 270: 0.412 to 2.25 urn → 0.412 to 2.25 μm

Row 275: Sentinel-3A → Sentinel 3A

Row 297: at the wavelengths 412 nm → at  412 nm

Row 298: channel → band

Row 349: to 0.018 → to 0.018 nm-1

Row 360: bbp_ (555) →

Row 372: Lw (555 nm) →

Row 406: AOT at 869 nm(aerosol optical → AOT at 869 nm (aerosol optical

Row 460: 0.15 m-1 → 0.15 m-1

Row 465: 0.0074 m−1  0.0074 m−1

Row 522: channel → band

Row 544: resources,A.S. and E.K..; → resources,A.S. and E.K.;

Author Response

(The authors gave the same response as above.)

Reviewer 3 Report

The author analyzed in detail the variability of the "blue" color index of in situ data in the Black Sea, and used a new regional algorithm to compare the retrieval of satellite values of Rrs (λ) in the short-wavelength region and field measurements of Black Sea. The proposed method has been proved that the "blue" color index varies very slightly in the Black Sea. And the "blue" color index can be used as a reference value in atmospheric correction algorithms. However, in the process of data analysis, the author did not better explain the innovation of his work and how to improve the accuracy of satellite reflectance in actual use. I recommend this work for publication after some revisions:

(1)   The description of Blue Index in the introduction should be placed in the method.

(2 On page 3, Line 112, the author should give the full name of “CI” because it first appeared.

(3) The author should explain the novel contributions of this work in the introduction, as well as the similarities and differences of this work with previous publications.

(4 The author compared three types of satellite sensors: MODIS Aqua/Terra, VIIRS SNPP, and Sentinel OLCI. What are the advantages of adapting these sensors over others in this case? How will this affect the results?

(5) What is the basis for selecting rectangle of pixels (5 × 5) in line 368 of page 9?

(6 In line 323 of page 9, the author proposed a linear regression for CI (412/443) and Rrs (412). The authors can use the scatterplot fit results of this linear relationship to demonstrate the validity of the conclusion.

(7) What is the role of the functional relationship between AOT and Rrs shown by the authors in Figure 5(b)? Please explain in detail.

(8) The discussion is weak. The author's discussion seems to be a further application of the research results, which may confuse the reader if it is a new research result. The discussion should briefly summarize the main implications of the findings, limitations, and recommendations for future research.

Author Response

(The authors gave the same response as above.)

Round 2

Reviewer 3 Report

Agree to publish in present form.

Author Response

Dear Editor and Reviewer!

Thank you very much for your response.

I apologize for the fact that some points of the answer for the first reviewer were missed or incorrectly formulated. We tried to analyze each of your remarks and, having answered it, supplement it in the text of the manuscript. I hope that the new point-to-point answers (see below in attached file) will reveal the essence of the questions.

We also made changes to the english language and style are fine/minor spell check that were required.

I hope these answers are valid. All other points and comments of the first reviewer are reflected in detail in the first response to the reviewer and the updated text of the manuscript. For further questions about our methodology, do not hesitate to contact me.

Kind Regards,

Evgeny Shybanov
